# Plant Natural Sources of the Endocannabinoid (*E*)-β-Caryophyllene: A Systematic Quantitative Analysis of Published Literature

**DOI:** 10.3390/ijms21186540

**Published:** 2020-09-07

**Authors:** Massimo E. Maffei

**Affiliations:** Department of Life Sciences and Systems Biology, University of Turin, Via Quarello 15/a, 10135 Turin, Italy; massimo.maffei@unito.it; Tel.: +39-011-670-5967

**Keywords:** plant species, essential oil, yield, percentages of (E)-β-caryophyllene, Belfrit list, plant part, geographical origin

## Abstract

(E)-β-caryophyllene (BCP) is a natural sesquiterpene hydrocarbon present in hundreds of plant species. BCP possesses several important pharmacological activities, ranging from pain treatment to neurological and metabolic disorders. These are mainly due to its ability to interact with the cannabinoid receptor 2 (CB2) and the complete lack of interaction with the brain CB1. A systematic analysis of plant species with essential oils containing a BCP percentage > 10% provided almost 300 entries with species belonging to 51 families. The essential oils were found to be extracted from 13 plant parts and samples originated from 56 countries worldwide. Statistical analyses included the evaluation of variability in BCP% and yield% as well as the statistical linkage between families, plant parts and countries of origin by cluster analysis. Identified species were also grouped according to their presence in the Belfrit list. The survey evidences the importance of essential oil yield evaluation in support of the chemical analysis. The results provide a comprehensive picture of the species with the highest BCP and yield percentages.

## 1. Introduction

The endogenous cannabinoid system (ECS) plays an important role in the immune response to an infection. At present, two cannabinoid (CB) receptors are described: cannabinoid type 1 receptor (CB1) and cannabinoid type 2 receptor (CB2), both G-protein coupled receptors [1]. The CB2 receptor represents the peripheral CB, due to its expression on circulating immune cells. However, studies have also found CB2 expression in the brain, such as cerebellum and microglial cells [2]. The CB2 receptor is involved in the attenuation of inflammatory immune responses. CB2 receptor pathway activation entails the suppression of cytokine release from immune cells and thereby dampening of the inflammatory response (immunosuppression) [3].

(*E*)-β-caryophyllene (BCP) is a bicyclic sesquiterpene hydrocarbon which is present in the essential oil of several plant species [4]. The Research Institute for Fragrance Materials (RIFM) evaluated BCP safety and the molecule has been approved by the Food and Drug Administration and by the European Food Safety Authority as a flavoring agent, which can be used in cosmetic and food additives [5]. Reports on oral sub-chronic toxicity support the safety of BCP for its proposed use also in medical food products [5]. BCP has been reported to be active against several disorders, with particular reference to cancer, chronic pain and inflammation [2]. Non-clinical BCP toxicity and an absence of adverse effects have been described [6]. Moreover, BCP can act as a selective agonist of CB2 [1], it activates peroxisome proliferator-activated receptor-α (PPAR α) [7] and has been recently involved in the prevention of lipid accumulation and in the improvement of glucose uptake [8]. Therefore, BCP is a plant-derived bioactive molecule able to improve health and prevent lifestyle diseases. Moreover, the specificity of BCP for the CB2 receptor, mainly expressed in peripheral tissues, and its inability to bind CB1, which is predominantly expressed at the level of the central nervous system, implies that its action is devoid of the known psychoactive effects associated with the activation of CB1 [1,2,9,10]. In this context, BCP is an interesting alternative to the use of Cannabis.

Owing to the growing importance of BCP, it was interesting to evaluate the occurrence of this important endocannabinoid in plant species used for the extraction of essential oils. Therefore, the aim of this work was to look for plant natural sources of BCP in order to provide the pharmaceutical, nutraceutical and aroma industries a summary of plant species, parts used for extraction and geographical origin of plants producing BCP. Moreover, additional information was provided with regards to the content and yield of BCP as well as the occurrence of selected species in the Belfrit list [11], which includes botanicals allowed in food supplements and ensures compliance of botanicals in terms of quality and safety.

## 2. Results and Discussion

The database search (performed in July 2020) for the term caryophyllene provided 5867 entries. The search was then refined by selecting all papers with a chemical composition description. This selection provided 2604 entries, which were individually analyzed in order to select papers providing information on BCP percentage > 10%. Papers were then analyzed and the species binomial name, the plant family, the country of origin of samples and the plant part extracted were reported along with the BCP percentage and yield percentage. The total number of selected species was 295 (Table 1). Table 1 also lists the presence of the species in the Belfrit list [11].

In general, the 295 species belonged to 51 families and were reported from 56 countries worldwide. The essential oil containing BCP was extracted from 13 different plant parts. Out of 295 species, 34 were found to be listed in the Belfrit list, whereas for 51 species no data were available on the yield percentage. In many cases, the researchers used a small amount of plant parts (ranging from a few g to 200–300 g) from which it was impossible to evaluate the oil yield. However, in the majority of the other cases the yield was provided and hence reported (Table 1).

The essential oil yield of 243 species ranged from 0.001 to 8.58%, whereas the BCP percentage of all selected species ranged from 9.8 (the threshold minimum level for species selection) to 75.6% (Table 2), providing an average percentage of 0.42% for yield and 27.4% for BCP. As shown in Table 2, variability was higher for yield percentages than for BCP percentage. The reason for the yield and BCP variability depends on several factors, including plant part, the quantity of plant material distilled and, most of all, the genetic variability and phenotypic plasticity of plants [303,304,305,306].

In order to look for plant species with the highest BCP and yield percentages, a scatter plot was obtained, as depicted in Figure 1. The highest yield and BCP percentages were found for *Copaifera langsdorffii*. High BCP percentages but with decreasing yields were found for *Bursera microphylla*, *Scutellaria havanensis* and *Pentadesma butyracea*. *Copaifera* species, popularly known as copaiba oil, are widely used in Brazilian popular medicine and the genus is known for its high essential oil yield and BCP content [135,307,308]. The genus *Bursera* belongs to the plant family Burseraceae and contains several aromatic spices producing oleo-gum resins, such as the traditional incenses, frankincense and myrrh [309]. *Pentadesma butyracea* (Clusiaceae) is a dense forest species which is found in the center and north of Benin forests whose bark, rough and deeply cracked, exudes a thick resinous juice, of reddish yellow color [115]. The *Scutellaria* genus (Lamiaceae) consists of plants which are widely distributed throughout the world; *S. butyracea* is an endemic plant native from Havana and is ethnomedically used for several purposes because of its BCP content [196].

High yields with lower BCP percentages were found for *Acalypha fruticosa, Achyrocline alata, Agrimonia eupatoria, Bowdichia virgilioides, Bursera microphylla, Croton pulegiodorus, Curcuma longa, Glechon marifolia, Laser trilobum, Meristotropis xanthioides, Origanum majorana, Pimpinella kotschyana, Piper guineense, Rosa canina, Salvia canariensis, Spondias pinnata, Syzygium aromaticum and Thuja orientalis*. All other species had a yield ranging from 0.004 to 1% and a BCP content ranging from 9.8 to 55 % (Figure 1).

The plant part that contained the highest content of BCP was then analyzed. In order to evidence the statistical linkage between the plant parts, a cluster analysis was calculated by considering as category the plant part and as variables the number of species, the BCP% and the yield% reported in Table 1 (Figure 2). Euclidean distances were calculated by using the average linkage method. Five clusters were evidenced: the first cluster was made by plant parts reported in more than 100 species and was dominated by leaves and aerial parts, which contained in general a BCP percentage lower than 28%. The other four clusters were made by plant parts reported in less than 16 species. These four clusters were further subdivided according to their BCP content (Figure 2). As expected, the highest BCP percentage was found in oleo-gum resins (cluster 2), followed by roots, barks and branches (cluster 3). Flowers and buds (cluster 4) showed a high yield, whereas twigs and woods (cluster 5) had both low yields and BCP percentages (Figure 2).

Table 3 summarizes the statistical analysis of BCP and yield percentages reported from different plant parts.

The next analysis was at the familial level. A cluster analysis was calculated with average linkage method by using data of Table 1 by considering as a category the plant families and the species number, yield% and BCP% as variables. The results of the cluster analysis show the presence of 6 clusters (Figure 3). The first cluster is made by the Asteraceae and the Lamiaceae which consist of a number of species > 50 and a BCP% < 31. The second cluster gathers all families whose species have a BCP% > 35%; in this cluster, the Magnoliaceae and the Papilionaceae are separated in a subcluster because of their high BCP% and low yield%, whereas the Fabaceae (which include the above mentioned *C. langsdorffii*) are separated in a subcluster because of their high yield %. The third cluster is made by families with a number of species > 13 and a BCP% > 23%; here, the Lauraceae, the Apiaceae and the Zingiberaceae are separated in a subcluster because of their higher BCP%. The genus *Ocotea* is one of the largest of the Lauraceae family, with approximately 350 species distributed throughout tropical and subtropical America. *O. splendens*, as many other *Ocotea* species [212] is characterized by a high percentage of BCP [217]. In the Apiaceae family, the species *P. kotschyana* spreads widely through Anatoly, Iran (northwest, west and center) and north of Iraq and contains BCP in all plant parts [41]. The family Zingiberaceae is well known for producing essential oils that are used to prevent and control several diseases; the species *R. breviscapa* was found to possess a high percentage of BCP [300]. The fourth cluster is made by families with a BCP% > 26 and a subcluster separates the Atherospermaceae, the Flacourtiaceae and the Meliaceae because of their BCP%. The fifth cluster is made by families with a BCP% < 25 and the Plantaginaceae are separated in a subcluster because of their relatively higher yield%. Finally, the sixth cluster is made by plant families with a low BCP percentage and a subcluster separates the Hernandiaceae, the Juglandaceae, the Phyllanthace and the Ptaeroxylaceae because of their BCP content lower than 11%.

Table 4 describes the statistical data related to plant families.

The next analysis aimed to evidence the geographical areas from which the plant species listed in Table 1 were collected. A cluster analysis was calculated with average linkage method, considering the country of origin as a category of their species number, yield% and BCP% as variables. The results of the cluster analysis show the presence of 6 clusters (Figure 4). The first cluster gathers countries with the highest number of species and a BCP percentage higher than 28%; here, a subcluster separates Brazil from India and Iran because of the higher number of species, in agreement with the literature data [310]. The second and third clusters identify countries where BCP has the highest percentages, whereas the fourth cluster gathers countries with a number of species higher than 8. The fifth cluster is made by countries where the BCP content is the lowest, whereas the sixth cluster is made by two subclusters with BCP percentages ranging from 18 to 25%. One of these subclusters is made by countries (Colombia, Fiji, Kenya, Morocco, Niger, North Korea, Portugal and Togo) where the species had a BCP percentage higher than 24% (Figure 4).

Table 5 summarizes the statistics related to countries of origin.

In order to separate which species containing BCP were also represent in the Belfrit list, a scatter plot was obtained by selecting BCP% and yield% as variables (Figure 5). *C. langdorffii*, *S. aromaticum*, *C. longa* and *B. virgilioides* were characterized by a yield ranging from 2 to 28%, with varying percentages of BCP; on the other hand, high percentages of BCP but lower yields% were found for *A. eupatoria*, *H. coronarium*, *C. odorata*, *P. americana* and *M. keonigi*. All other species showed both lower yields and BCP percentage.

## 3. Materials and Methods

### 3.1. Systematic Analysis of BCP-Containing Plant Species

After a preliminary search by using different databases, the work was performed by using Clarivate Analytics Web of Science as a database (http://apps.webofknowledge.com). The basic search criterion was on the general search for the molecule (caryophyllene), then the exclusion criteria were the presence of BCP and a percentage of BCP in the reported results higher than 10%. Papers reporting the occurrence of BCP where then downloaded and saved as a pdf for further reading and collection of information.

### 3.2. Statistical Analysis

The binomial name of the species (including the author), the family of belonging, the plant part used, the country of origin of the sample, the yield and the BCP percentages were inserted in a database by using Systat^®^ 10 software (Systat Software Inc., San Jose, California, U.S.A.). Data were organized in columns and used for further processing. Average values along with ranges, standard deviation (S.D.), standard error of the mean (S.E.M.) and coefficient of variation (C.V.) were calculated by considering as grouping categories either the species, families, country of origin or plant part used. As a classification statistical method, a cluster analysis was calculated by considering for each category the total number of species, the BCP percentage and the yield percentage by using Systat^®^ 10 software. Euclidean distances were calculated with the average linkage method. Data were plotted as either scatter plots of yield percentage vs. BCP percentage or dendrograms showing the different clusters according to the calculated distance.

## 4. Conclusions

The attractiveness of BCP, a natural sesquiterpene present in the essential oil of different plant species, arises from its pharmacological feature as a CB2 receptor agonist. This characteristic, along with the lack of interaction with the CB1, makes BCP an interesting plant endocannabinoid with the advantage of lacking any psychotropic effect, as is typical of some Cannabis extracts [8,311,312].

This systematic analysis of published literature on plant species containing BCP in their essential oils identified the species with the highest yield and BCP content and allowed to select which species are also present in the Belfrit list (i.e., potentially attractive for pharmaceutical and nutraceutical industries).

This survey also evidenced the common practice of many authors to ignore the importance of providing the yield of the distilled essential oil, which represent a basic starting point for all industrial applications of the plant species under study. This problem was often correlated with the low amount of plant material distilled. Although interesting from a chemical-analytical point of view, the sole chemical analysis of the essential oil is not useful if performed on a single plant or a few plants, because it does not provide any information on the population genetic variability, being mainly affected by phenotypic plasticity, which is responsible for individual variations inside a population [305].

This work identified some top species like *C. langsdforffii*, *C. odorata*, *H. lupulus*, *P. nigrum* and *S. aromaticum*, which provide a high percentage of BCP along with interesting yields. These species, upon a skillful molecular fractionation to remove undesired/toxic monoterpenes, may provide high percentages of BCP that can be used for the preparation of new drugs or dietary supplements aimed to improve health, prevent lifestyle diseases and act as a valid support for chronical diseases such as pain, metabolic and neurological disorders.

## Figures and Tables

**Figure 1 ijms-21-06540-f001:**
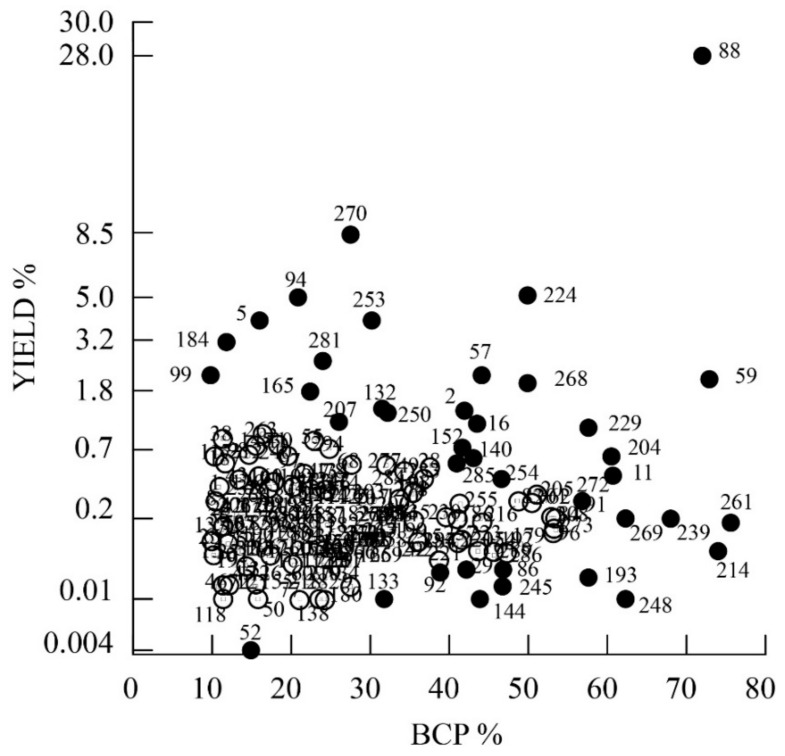
Scatter plot of BCP percentage vs. yield percentage. The yield axis is represented as a power of 0.3 scale in order to evidence species with yields ranging from 0.004 to 3%. Numbers correspond to plant species listed in Table 1. Filled circles outline the species outside the central group of all other species (hollow circles).

**Figure 2 ijms-21-06540-f002:**
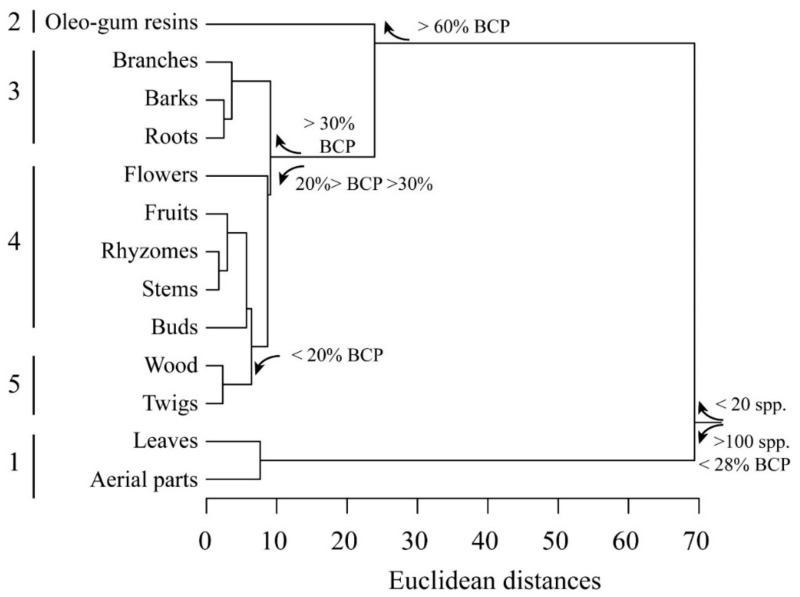
Cluster analysis of BCP and yield percentages according to the plant part used for extraction. Euclidean distances are calculated with average linkage method. Five clusters are evident (see text for explanation).

**Figure 3 ijms-21-06540-f003:**
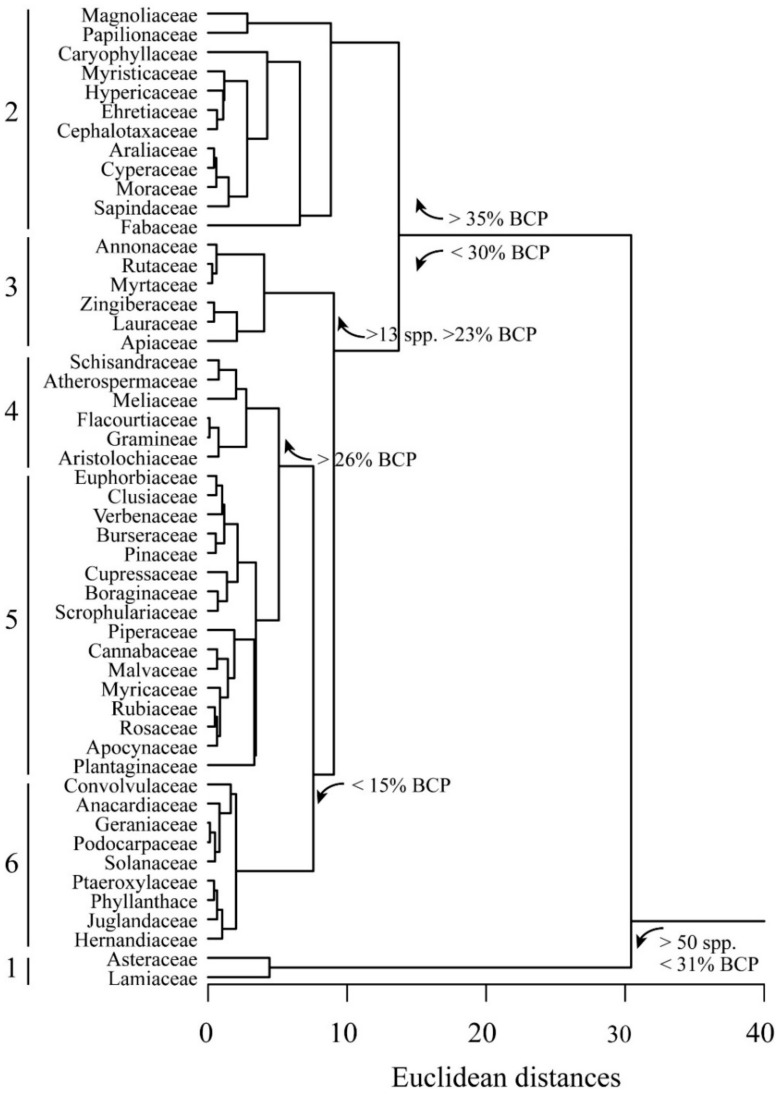
Cluster analysis of BCP and yield percentages according to the plant families. Euclidean distances are calculated with average linkage method. Six clusters are evident (see text for explanation).

**Figure 4 ijms-21-06540-f004:**
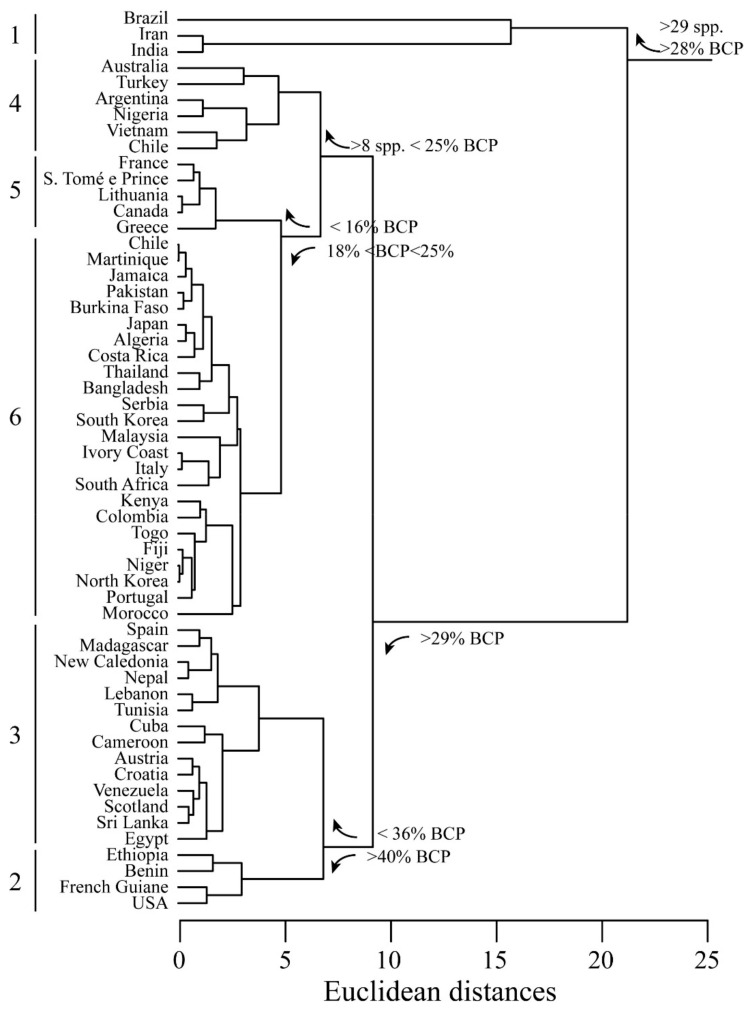
Cluster analysis of BCP and yield percentages according to the country of origin of extracts. Euclidean distances are calculated with average linkage method. Six clusters are evident (see text for explanation).

**Figure 5 ijms-21-06540-f005:**
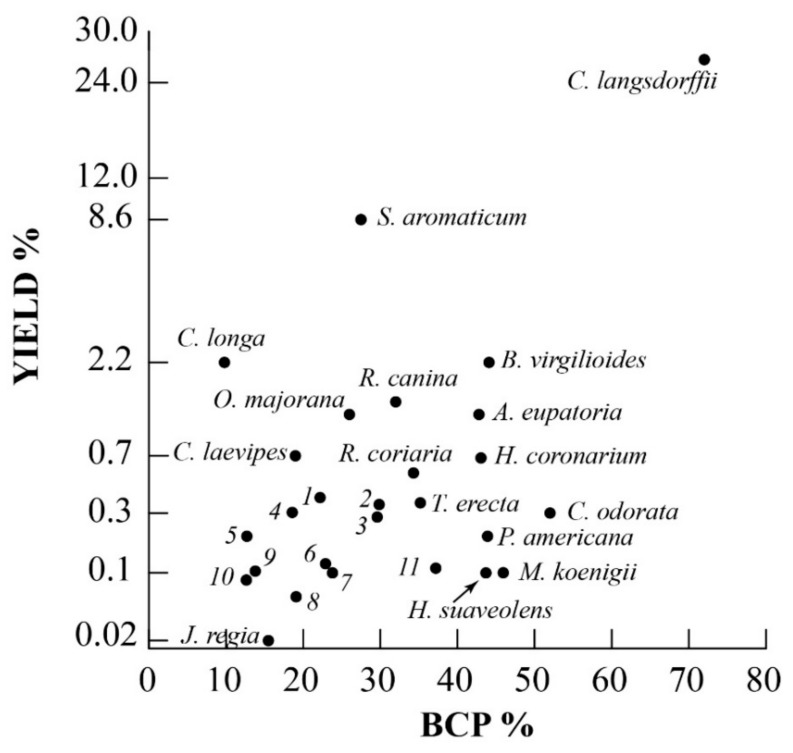
Scatter plot of BCP% and yield% of plant species present in the Belfrit list. The yield axis is scaled as a power of 0.2 in order to evidence species with yields ranging from 0.02 to 8.6%. 1, *Pinus pinaster* Aiton; 2, *Ocimum tenuiflorum* L.; 3, *Aegle marmelos* (L.) Corr.; 4, *Protium heptaphyllum* (Aubl.) March.; 5, *Artemisia verlotiorum* Lam *rinus officinalis* L.; 6, *Annona squamosa* L.; 7, *Cannabis sativa* L.; 8, *Centella asiatica* L.; 9, *Annona muricata* L.; 10, *Rosmarinus officinalis* L; 11, *Perilla frutescens var. japonica* (Hassk.) H. Hara.

**Table 1 ijms-21-06540-t001:** Occurrence of (*E*)-β-caryophyllene (BCP) in different plant species. n.a., data not available, the essential oil (E.O.) yield is expressed as volume/weight percentage.

Family	Genus	Species and Auth	Geogr. Origin of Sample	Belfrit List	Part Used	E.O. Yield%	BCP%	Code	Ref.
Anacardiaceae	*Rhus*	*coriaria* L.	Iran	YES	fruits	0.55	34.3	249	[12]
Anacardiaceae	*Spondias*	*pinnata* (Linn. F.) Kurz	Egypt	NO	leaves	2.00	49.9	268	[13]
Annonaceae	*Annona*	*muricata* L.	Bénin	YES	leaves	0.10	13.6	30	[14]
Annonaceae	*Annona*	*densicoma* Mart.	Brazil	NO	leaves	0.10	14.4	31	[15]
Annonaceae	*Annona*	*senegalensis* Pers.	Burkina Faso	NO	leaves	0.73	19.1	32	[16]
Annonaceae	*Annona*	*squamosa* L.	India	YES	leaves	0.12	22.9	33	[17]
Annonaceae	*Artabotrys*	*hexapetalus* (L. f.) Bhandare	Vietnam	NO	flowers	0.94	11.4	38	[18]
Annonaceae	*Cananga*	*odorata* (Lam.) Hook.f. and Thomson	Australia	YES	leaves	0.30	52.0	62	[19]
Annonaceae	*Cleistopholis*	*glauca* Pierre ex Engler and Diels	Ivory Coast	NO	leaves	0.19	26.2	81	[20]
Annonaceae	*Fissistigma*	*rubiginosum* Merr.	Vietnam	NO	leaves	0.30	28.1	125	[21]
Annonaceae	*Goniothalamus*	*multiovulatus* Ast	Vietnam	NO	stems	0.21	35.7	135	[22]
Annonaceae	*Melodorum*	*sp.* (Dunal) Hook.f. and Thomson	Australia	NO	leaf	0.15	26.7	182	[23]
Annonaceae	*Miliusa*	*horsfieldii* (Bennett) Baillon ex Pierre	Australia	NO	leaves	0.1	20.2	188	[24]
Annonaceae	*Mitrephora*	*zippeliana* Miq.	Australia	NO	leaves	0.30	18.1	189	[19]
Annonaceae	*Polyalthia*	*oliveri* Engl.	Ivory Coast	NO	leaves	0.13	31.4	237	[25]
Annonaceae	*Pseuduvaria*	*hylandii* Jessup	Australia	NO	leaves	0.50	24.1	242	[26]
Annonaceae	*Uvariodendron*	*calophyllum* R. E. Fries	Cameroon	NO	stem barks	0.52	32.5	284	[27]
Apiaceae	*Berula*	*erecta* (Hudson) Coville *subsp. erecta*	Serbia	NO	aerial parts	0.01	14.9	52	[28]
Apiaceae	*Bilacunaria*	*anatolica* A. Duran	Turkey	NO	aerial parts	0.14	10.3	54	[29]
Apiaceae	*Centella*	*asiatica* L.	South Africa	YES	aerial parts	0.06	19.1	75	[30]
Apiaceae	*Conium*	*maculatum* L.	Iran	NO	aerial parts	0.20	15.3	85	[31]
Apiaceae	*Dorema*	*aucheri* Boiss.	Iran	NO	leaves	0.40	35.7	108	[32]
Apiaceae	*Eryngium*	*vesiculosum* Labill.	Australia	NO	aerial parts	n.a.	20.3	116	[33]
Apiaceae	*Ferula*	*glauca* L.	Iran	NO	leaves	0.07	24.9	123	[34]
Apiaceae	*Grammosciadium*	*pterocarpum* Boiss.	Turkey	NO	aerial parts	n.a.	15.3	136	[35]
Apiaceae	*Hippomarathrum*	*microcarpum* (M. Bieb.) B. Fedtsch	Iran	NO	aerial parts	0.85	15.75	145	[36]
Apiaceae	*Hippomarathrum*	*boissieri* Reuter et Hausskn	Turkey	NO	aerial parts	0.40	25.6	146	[37]
Apiaceae	*Laser*	*trilobum* (L.) Borkh.	Iran	NO	aerial parts	1.80	22.4	165	[38]
Apiaceae	*Oenanthe*	*divaricata* (R. Br.) Mabb.	Spain	NO	aerial parts	0.20	15.3	206	[39]
Apiaceae	*Ostericum*	*viridiflorum* (Turcz.) Kitagawa	China	NO	aerial parts	0.03	24.3	210	[40]
Apiaceae	*Pimpinella*	*kotschyana Boiss.*	Iran	NO	seeds	5.16	49.9	224	[41]
Apiaceae	*Prangos*	*uloptera* DC.	Iran	NO	aerial parts	0.70	18.2	240	[42]
Apiaceae	*Zosima*	*absinthifolia* Link	Iran	NO	aerial parts	0.20	22.2	295	[43]
Apocynaceae	*Allamanda*	*cathartica* L.	Brazil	NO	flowers	n.a.	15.7	21	[44]
Apocynaceae	*Aspidosperma*	*cylindrocarpon* Muell. Arg.	Brazil	NO	leaves	0.03	14.3	45	[45]
Apocynaceae	*Tabernaemontana*	*catharinensis* A. DC.	Brazil	NO	leaves	0.30	56.9	272	[46]
Araliaceae	*Schefflera*	*stellata* (Gaertn.) Harms	India	NO	leaves	0.10	19.2	260	[47]
Aristolochiaceae	*Aristolochia*	*elegans* Mast.	Argentina	NO	leaves	n.a.	27.8	36	[48]
Aristolochiaceae	*Aristolochia*	*fordiana* Hemsl	China	NO	aerial parts	0.19	11.1	37	[49]
Asteraceae	*Achillea*	*asplenifolia* Vent.	Serbia	NO	aerial parts	0.10	17.6	4	[50]
Asteraceae	*Achyrocline*	*alata* (D.C.)	Brazil	NO	leaf and flowers	4.00	16.0	5	[51]
Asteraceae	*Acroptilon*	*repens* (L.)	Iran	NO	aerial parts	0.11	10.0	6	[52]
Asteraceae	*Ageratum*	*fastigiatum* (Gardn.) R. M. King et H. Rob	Brazil	NO	branches	0.20	34.9	13	[53]
Asteraceae	*Ageratum*	*conyzoides* L.	Portugal	NO	flowers	0.17	24.6	14	[54]
Asteraceae	*Anthemis*	*altissima* L.	Iran	NO	flowers	0.03	25.3	34	[55]
Asteraceae	*Artemisia*	*verlotiorum* Lamotte	France	YES	aerial parts	0.20	12.7	39	[56]
Asteraceae	*Artemisia*	*parviflora* Roxb	India	NO	aerial parts	0.20	15.3	40	[57]
Asteraceae	*Artemisia*	*roxburghiana* Besser *var. purpurascens* (Jacq.) Hook	India	NO	aerial parts	0.85	18.4	41	[58]
Asteraceae	*Artemisia*	*capillaris* Thunb	South Korea	YES	aerial parts	n.a.	11.1	42	[59]
Asteraceae	*Artemisia.*	*stricta* Edgew. *f. stricta* Pamp	India	NO	aerial parts	0.46	13.4	43	[60]
Asteraceae	*Artemisia.*	*lavandulaefolia* DC	South Korea	NO	aerial parts	n.a.	16.1	44	[61]
Asteraceae	*Aspilia*	*africana* (Pers.) C. D. Adams	Nigeria	NO	leaves	0.02	10.8	46	[62]
Asteraceae	*Baccharis.*	*articulata* (Lam.) Pers	Argentina	NO	aerial parts	n.a.	16.8	48	[63]
Asteraceae	*Bidens*	*pilosa* L.	Cameroon	NO	leaves	n.a.	27.1	53	[64]
Asteraceae	*Centaurea*	*zlatiborensis* Zlatkovic, Novakovic and Janackovic	Serbia	NO	flowers	n.a.	28.3	73	[65]
Asteraceae	*Centaurea*	*appendicigera* C. Koch	Turkey	NO	aerial parts	0.18	17.5	74	[66]
Asteraceae	*Centratherum*	*punctatum* Cass	Nigeria	NO	leaves	n.a.	16.6	76	[67]
Asteraceae	*Chromolaena*	*odorata* L.	Togo	NO	aerial parts	0.50	25.2	78	[68]
Asteraceae	*Conyza*	*bonariensis* (L.) Cronquist	Brazil	NO	aerial parts	0.20	14.4	87	[69]
Asteraceae	*Cyanthillium*	*cinereum* (L.) H. Rob	Ivory Coast	NO	roots	n.a.	17.0	100	[70]
Asteraceae	*Dendranthema*	*indicum* (L.) Des Moul.	China	NO	aerial parts	0.08	13.8	106	[71]
Asteraceae	*Emilia*	*sonchifolia* (L.) DC.	India	NO	aerial parts	n.a.	22.7	110	[72]
Asteraceae	*Epaltes*	*alata* Steetz	Niger	NO	leaves	0.30	24.0	111	[73]
Asteraceae	*Eremanthus*	*erythropappus* (DC.) MacLeish	Brazil	NO	leaves	0.12	29.3	113	[74]
Asteraceae	*Erigeron*	*ramosus* (Walt.) B.S.P.	Korea	NO	flowers	0.40	24.0	114	[75]
Asteraceae	*Eriocephalus*	*luederitzianus* O.Hoffm.	South Africa	NO	aerial parts	0.10	13.3	115	[76]
Asteraceae	*Eupatorium*	*triplinerve* Vahl	India	NO	leaves	0.40	14.7	120	[77]
Asteraceae	*Flourensia*	*campestris*	Argentina	NO	aerial parts	0.02	15.3	127	[78]
Asteraceae	*Helichrysum*	*indutum* Humbert	Madagascar	NO	aerial parts	0.19	33.1	141	[79]
Asteraceae	*Helichrysum*	*kraussii* Sch. Bip.	South Africa	NO	aerial parts	n.a.	30.7	142	[80]
Asteraceae	*Helichrysum*	*melaleucum* Rchb. ex Holl.	Spain	NO	aerial parts	0.10	35.4	143	[39]
Asteraceae	*Koanophyllon*	*villosum* (Sw.) King et Robins	Cuba	NO	aerial parts	0.45	17.0	160	[81]
Asteraceae	*Laggera*	*oloptera* (DC.) C. D. Adams	Cameroon	NO	leaves	0.05	20.4	161	[82]
Asteraceae	*Microglossa*	*pyrrhapappa var. pyrrhopappa* (A. Rich) Agnew	Kenya	NO	leaves	0.40	20.3	185	[83]
Asteraceae	*Mikania*	*cordata* (Burm.f.) B.L. Robinson *var. cordata*	Ivory Coast	NO	leaves	0.63	11.8	187	[84]
Asteraceae	*Oyedaea*	*verbesinoides* DC.	Venezuela	NO	leaves	0.05	27.1	211	[85]
Asteraceae	*Perymenium*	*grande* Hemsl. *var. nelsonii* (Robins. and Greenm.) Fay	Costa Rica	NO	leaves	0.30	30.5	217	[86]
Asteraceae	*Petasites*	*japonicus* (Siebold and Zucc.) Maxim.	Japan	NO	leaves	0.02	21.9	218	[87]
Asteraceae	*Pluchea*	*carolinensis* (Jacq.) Sweet	Martinique	NO	leaves	0.11	21.1	236	[88]
Asteraceae	*Porophyllum*	*obscurum* (Spreng.) D.C.	Argentina	NO	leaves	0.30	14.1	238	[89]
Asteraceae	*Solidago*	*decurrens* Lour	China	NO	leaves	0.37	15.4	266	[90]
Asteraceae	*Tagetes*	*patula* L.	Austria	NO	flowers	0.15	53.5	273	[91]
Asteraceae	*Tagetes*	*erecta* L.	Iran	YES	flowers	0.35	35.2	274	[92]
Asteraceae	*Tanacetum*	*punctatum* (Desr.) Grierson	Iran	NO	aerial parts	0.1	21.1	275	[93]
Asteraceae	*Tarchonanthus*	*trilobus var. galpinii* (Hutch. and E.Phillips) Paiva	South Africa	NO	leaves	0.14	30.4	276	[94]
Asteraceae	*Vernonia*	*chalybaea* Mart.	Brazil	NO	aerial parts	0.10	39.1	287	[95]
Asteraceae	*Vernonia*	*scorpioides* (Lam.) Pers.	Brazil	NO	aerial parts	0.10	30.6	288	[96]
Asteraceae	*Xanthium*	*strumarium* L.	Pakistan	NO	leaves	n.a.	17.5	291	[97]
Asteraceae,	*Leptocarpha*	*rivularis* DC.	Chile	NO	aerial parts	0.15	21.1	168	[98]
Atherospermataceae	*Daphnandra*	*repandula* (F.Muell.) F.Muell.	Australia	NO	aerial parts	0.20	12.2	105	[99]
Boraginaceae	*Cordia*	*leucocephala* Moric	Brazil	NO	leaves	0.04	39.0	91	[100]
Boraginaceae	*Cordia*	*multispicata* Cham.	Brazil	NO	leaves	0.25	56.6	92	[101]
Burseraceae	*Bursera*	*aromatica* (Proctor)	Jamaica	NO	leaves	0.03	21.7	59	[102]
Burseraceae	*Bursera*	*microphylla* A. Gray	USA	NO	oleo-gum-resin	2.10	72.9	60	[103]
Burseraceae	*Canarium*	*parvum* Leen.	Vietnam	NO	leaves	0.20	18.7	63	[104]
Burseraceae	*Dacryodes*	*edulis* (G. Don) H. J. Lam	Nigeria	NO	leaves	0.08	26.0	103	[105]
Burseraceae	*Protium*	*heptaphyllum* (Aubl.) March.	Brazil	YES	leaves	0.30	18.6	241	[106]
Cannabaceae	*Cannabis*	*sativa* L. *ssp. spontanea*	Austria	YES	aerial parts	n.a.	16.2	64	[107]
Cannabaceae	*Cannabis*	*sativa* L.	Italy	YES	flowers	0.10	23.8	65	[108]
Cannabaceae	*Humulus*	*lupulus* L.	USA	YES	aerial parts	n.a.	22.0	148	[109]
Caryophyllaceae	*Dianthus*	*caryophyllus* L.	Iran	YES	aerial parts	n.a.	34.8	107	[110]
Cephalotaxaceae	*Cephalotaxus*	*harringtonia* K.Koch *subsp. harringtonia*	India	NO	twigs	0.01	21.1	77	[111]
Clusiaceae	*Clusia*	*nemorosa* G. Mey	Brazil	NO	fruits	0.30	48.6	83	[112]
Clusiaceae	*Garcinia*	*atroviridis* Griff. ex T. Anders.	Malaysia	NO	fruits	n.a.	23.8	128	[113]
Clusiaceae	*Kielmeyera*	*rugosa* Choisy	Brazil	NO	fruits	n.a.	16.4	158	[114]
Clusiaceae	*Pentadesma*	*butyracea* Sabine	Benin	NO	barks	0.08	74.0	214	[115]
Clusiaceae	*Psorospermum*	*corymbiferum* Hochr	Nigeria	NO	leaves	0.02	46.8	245	[116]
Convolvulaceae	*Convolvulus*	*persicus* L.	Iran	NO	aerial parts	0.04	47.0	86	[117]
Cupressaceae	*Cedrus*	*atlantica* G. Manetti	Algeria	NO	twigs	0.02	11.4	72	[118]
Cupressaceae	*Juniperus*	*macrocarpa* Sibth. and Sm. (Jom)	Turkey	NO	fruits	n.a.	29.6	156	[119]
Cupressaceae	*Thuja*	*orientalis* L.	Egypt	NO	aerial parts	2.60	24.0	281	[120]
Cyperaceae	*Cyperus*	*glomeratus* L.	Serbia	NO	rhizomes and roots	0.06	12.6	102	[121]
Ehretiaceae	*Varronia*	*curassavica* Jacq.	Brazil	NO	leaves	0.6	41.2	285	[122]
Ehretiaceae	*Varronia*	*schomburgkii* (DC.) Borhidi	French Guiana	NO	aerial parts	0.06	47.0	286	[123]
Euphorbiaceae	*Acalypha*	*fruticosa* Forssk	India	NO	leaves	1.40	42.0	2	[124]
Euphorbiaceae	*Alchornea*	*tiliifolia* (Benth.) Muell.	Vietnam	NO	aerial parts	n.a.	10.7	20	[125]
Euphorbiaceae	*Croton*	*rhamnifolioides* Pax and Hoffm	Brazil	NO	leaf	0.21	33.3	94	[126]
Euphorbiaceae	*Croton*	*glandulosus* L.	Brazil	NO	aerial parts	0.12	53.2	95	[127]
Euphorbiaceae	*Croton*	*pulegiodorus* Baill.	Brazil	NO	aerial parts	5.00	20.9	96	[128]
Euphorbiaceae	*Phyllanthus*	*muellerianus* (O. Kuntze) Exell	Nigeria	NO	leaves	0.12	41.9	223	[129]
Fabaceae	*Bauhinia*	*rufa* Steud.	Brazil	NO	leaves	0.01	15.8	50	[130]
Fabaceae	*Bowdichia*	*virgilioides* Kunt	Brazil	YES	seeds	2.20	44.1	57	[131]
Fabaceae	*Caesalpinia*	*decapetala* (Roth) Alston	Japan	NO	aerial parts	0.07	17.2	61	[132]
Fabaceae	*Copaifera*	*langsdorffii Desf.*	Brazil	YES	oleoresins	28.00	72.0	88	[133]
Fabaceae	*Copaifera*	*multijuga* Hayne	Brazil	NO	oleoresins	n.a.	57.5	89	[134]
Fabaceae	*Copaifera*	*reticulata* Ducke	Brazil	NO	oleoresins	n.a.	68.0	90	[135]
Fabaceae	*Dalea*	*carthagenensis* L.	Colombia	NO	leaves	0.15	20.7	104	[136]
Fabaceae	*Eperua*	*duckeana* Cowan	Brazil	NO	leaves	n.a.	31.8	112	[137]
Fabaceae	*Glycyrrhiza*	*triphylla* Fisch. and C.A.Mey	Iran	NO	aerial parts	0.50	25.4	134	[138]
Fabaceae	*Psoralea*	*bituminosa L*	Italy	NO	leaves	0.10	23.2	244	[139]
Fabaceae	*Rynchosia*	*minima* DC.	Kenya	NO	aerial parts	0.10	30.4	252	[140]
Flacourtiaceae	*Casearia*	*decandra* Jacq.	Brazil	NO	leaves	0.20	13.0	67	[141]
Flacourtiaceae	*Casearia*	*sylvestris* Swart.	Brazil	NO	leaves	0.60	27.5	68	[142]
Geraniaceae	*Geranium*	*wallichianum* D. Don ex Sweet	India	NO	aerial parts	n.a.	15.9	130	[143]
Gramineae	*Elyonurns*	*muticus* (Sprengel) O.Kuntze	Brazil	NO	leaves	0.45	17.9	109	[144]
Gramineae	*Melinis*	*minutiflora* P. Beauv	Kenya	NO	aerial parts	0.01	24.2	180	[145]
Hernandiaceae	*Hernandia*	*nymphaeifolia* (C.Presl) Kubitzki	Australia	NO	leaves	0.01	43.8	144	[146]
Hypericaceae	*Hypericum*	*brasiliense* Choisy	Brazil	NO	aerial parts	0.10	29.5	150	[147]
Hypericaceae	*Hypericum*	*perforatum* L.	Iran	YES	aerial parts	n.a.	25.05	151	[148]
Hypericaceae	*Vismia*	*baccifera subsp. dealbata* (Kunth) Ewan	Venezuela	NO	leaves	0.07	45.7	289	[149]
Juglandaceae	*Juglans*	*regia* L.	India	YES	leaves	0.02	15.5	155	[150]
Lamiaceae	*Aegiphila*	*lhotzkiana* Cham.	Brazil	NO	leaves	0.02	27.5	9	[151]
Lamiaceae	*Ajuga*	*parviflora* Benth.	India	NO	aerial parts	n.a.	22.4	18	[152]
Lamiaceae	*Ajuga*	*comata* Stapf.	Iran	NO	aerial parts	n.a.	30.9	19	[153]
Lamiaceae	*Ballota*	*nigra* L.	Algeria	YES	aerial parts	n.a.	24.6	49	[154]
Lamiaceae	*Clerodendrum*	*polycephalum* Baker	Nigeria	NO	leaves	0.16	28.9	82	[155]
Lamiaceae	*Colquhounia*	*coccinea* Wall.	India	NO	flower	0.20	53.2	84	[156]
Lamiaceae	*Cunila*	*incana* Benth.	Brazil	NO	aerial parts	0.72	11.3	98	[157]
Lamiaceae	*Cyclotrichium.*	*strussii* Bornm	Iran	NO	aerial parts	0.37	16.9	101	[158]
Lamiaceae	*Glechoma*	*hederacea* L.	Lithuania	NO	aerial parts	0.05	14.2	131	[159]
Lamiaceae	*Glechon*	*marifolia* Benth.	Brazil	NO	leaves	1.40	32.2	132	[160]
Lamiaceae	*Hoslundia*	*opposita* Vahl.	Ivory Coast	NO	leaves	0.04	24.8	147	[161]
Lamiaceae	*Hymenocrater*	*calycinus* (Boiss.) Benth.	Iran	NO	aerial parts	0.20	32.8	149	[162]
Lamiaceae	*Hyptidendron*	*canum* (Pohl ex Benth.) Harley	Brazil	NO	leaves	0.82	41.6	152	[163]
Lamiaceae	*Hyptis*	*mutabilis* (Rich.) Briq.	Argentina	NO	aerial parts	n.a.	59.4	153	[164]
Lamiaceae	*Hyptis*	*suaveolens* (L.) Poit.	Bénin	YES	fruits	0.10	43.7	154	[165]
Lamiaceae	*Lallenmantia*	*iberica* (M. Bieb.) Fisch and CA Mey	Turkey	NO	aerial parts	n.a.	18.3	162	[166]
Lamiaceae	*Leonotis*	*ocymifolia* (Burm.f.) M.Iwarsson	South Africa	NO	leaves	0.06	30.8	166	[167]
Lamiaceae	*Leonurus*	*sibiricus* L.	Argentina	NO	aerial parts	n.a.	35.2	167	[164]
Lamiaceae	*Leucas*	*aspera* (Willd.) Link	India	NO	aerial parts	0.30	34.2	169	[168]
Lamiaceae	*Leucas*	*indica* (L.) R.Br	India	NO	aerial parts	n.a.	51.1	170	[169]
Lamiaceae	*Marrubium*	*bourgaei subsp. caricum* P.H.Davis	Tunisia	NO	aerial parts	0.07	23.2	175	[170]
Lamiaceae	*Marsypianthes*	*chamnedrys* (Vahl) Kuntze	Brazil	NO	aerial parts	n.a.	15.1	176	[171]
Lamiaceae	*Melissa*	*romana* Miller	Italy	NO	aerial parts	0.30	15.8	181	[172]
Lamiaceae	*Mentha*	*longifolia* (L.) Hudson	Iran	NO	aerial parts	0.41	23.2	183	[173]
Lamiaceae	*Micromeria*	*myrtifolia* Boiss. and Hohen.	Turkey	NO	aerial parts	0.20	40.8	186	[174]
Lamiaceae	*Mosla*	*soochowensis* Matsuda	China	NO	aerial parts	0.05	12.8	191	[175]
Lamiaceae	*Nepeta*	*fissa* C.A. Mey	Iran	NO	aerial parts	0.25	33.1	200	[176]
Lamiaceae	*Nepeta*	*curviflora* Boiss.	Lebanon	NO	aerial parts	0.30	50.2	201	[177]
Lamiaceae	*Ocimum*	*tenuiflorum* L.	India	YES	aerial parts	0.33	30.0	203	[178]
Lamiaceae	*Origanum*	*majorana* L.	Algeria	YES	aerial parts	1.20	26.0	207	[179]
Lamiaceae	*Orthodon*	*dianfhera* Maxim.	Vietnam	NO	aerial parts	0.20	52.9	208	[180]
Lamiaceae	*Orthosiphon*	*pallidus* Royle, ex Benth	India	NO	aerial parts	n.a.	17.4	209	[181]
Lamiaceae	*Perilla*	*frutescens var. japonica* (Hassk.) H.Hara	China	YES	leaves	0.11	37.2	215	[182]
Lamiaceae	*Phlomis*	*crinita* Cav. *ssp. mauritanica* Munby	Tunisia	NO	aerial parts	0.10	40.8	220	[183]
Lamiaceae	*Phlomis*	*rigida* Labill.	Turkey	NO	aerial parts	0.05	38.7	221	[184]
Lamiaceae	*Platostoma*	*menthoides* (L.) A. J. Paton	Sri Lanka	NO	aerial parts	0.50	37.0	233	[185]
Lamiaceae	*Plectranthus*	*rugosus* Wall.	India	NO	leaves	n.a.	38.4	234	[186]
Lamiaceae	*Pycnostachys*	*eminii* Gürke	Ethiopia	NO	leaves	0.13	21.6	246	[187]
Lamiaceae	*Rosmarinus*	*officinalis* L	Lebanon	YES	aerial parts	0.09	12.9	251	[188]
Lamiaceae	*Salvia*	*palaefolia* Kunth	Colombia	NO	aerial parts	0.06	32.2	253	[189]
Lamiaceae	*Salvia*	*bracteata* Banks and Soland	Iran	NO	aerial parts	0.28	41.4	254	[190]
Lamiaceae	*Salvia*	*hydrangea* DC. ex Benth.	Iran	NO	aerial parts	0.20	33.4	255	[191]
Lamiaceae	*Salvia*	*nemorosa* L.	Iran	NO	aerial parts	0.12	41.6	256	[192]
Lamiaceae	*Salvia*	*virgata* Jacq.	Iran	NO	aerial parts	0.48	46.6	257	[193]
Lamiaceae	*Salvia*	*canariensis* L.	Spain	NO	aerial parts	4.00	30.2	258	[194]
Lamiaceae	*Salvia*	*montbretii* Benth.	Turkey	NO	aerial parts	0.10	32.8	259	[195]
Lamiaceae	*Scutellaria*	*havanensis* Jacq.	Cuba	NO	leaves	0.18	75.6	261	[196]
Lamiaceae	*Scutellaria*	*brevibracteata* Stapf. *subsp. pannosula*	Turkey	NO	aerial parts	n.a.	36.4	262	[197]
Lamiaceae	*Sideritis*	*clandestina subsp. peloponnesiaca* (Boiss. and Heldr.) Baden	Greece	NO	aerial parts	1.00	16.4	263	[198]
Lamiaceae	*Sideritis*	*phlomoides* Boiss. and Bal.	Turkey	NO	aerial parts	0.20	30.7	264	[199]
Lamiaceae	*Stachys*	*viticina Boiss.*	Turkey	NO	aerial parts	0.20	62.3	269	[200]
Lamiaceae	*Teucrium*	*arduini* L.	Croatia	NO	aerial parts	0.35	35.4	277	[201]
Lamiaceae	*Teucrium*	*flavum* L.	Iran	NO	leaves	0.20	30.7	278	[202]
Lamiaceae	*Teucrium*	*siculum* (Raf.) Guss.	Italy	NO	aerial parts	0.10	30.9	279	[203]
Lamiaceae	*Teucrium*	*turredanum* Losa and Rivas-Goday	Spain	NO	aerial parts	0.60	32.0	280	[204]
Lamiaceae	*Viticipremna*	*queenslandica* Munir	Australia	NO	leaves	n.a.	33.6	290	[205]
Lamiaceae	*Ziziphora*	*taurica* M.Bieb. *subsp. taurica*	Turkey	NO	aerial parts	0.80	24.8	294	[206]
Lauraceae	*Aiouea*	*costaricensis* (Mez) Kosterm.	Costa Rica	NO	leaf	0.10	12.0	17	[207]
Lauraceae	*Alseodaphne*	*peduncularis* Meisn	Malaysia	NO	leaves	n.a.	24.0	27	[208]
Lauraceae	*Aniba*	*riparia* (Nees) Mez	Brazil	NO	leaves	0.30	16.9	29	[209]
Lauraceae	*Beilschmiedia*	*penangiana* Gamble	Malaysia	NO	aerial parts	0.10	12.6	51	[210]
Lauraceae	*Cassytha*	*pubescens* R.Br.	Australia	NO	aerial parts	0.10	30.9	69	[211]
Lauraceae	*Cinnamomum*	*tamala* (Ham) Nees and Eberm.	Pakistan	NO	leaves	0.03	25.3	79	[212]
Lauraceae	*Litsea*	*helferi* Hook.f.	Vietnam	NO	leaves	0.30	14.2	172	[213]
Lauraceae	*Nectandra*	*lanceolata* Ness	Brazil	NO	leaves	0.20	32.5	198	[214]
Lauraceae	*Neolitsea*	*foliosa* (Nees) Gamble *var. caesia* (Meisner) Gamble	India	NO	leaves	0.10	35.3	199	[215]
Lauraceae	*Ocotea*	*duckei* Vattimo-Gil	Brazil	NO	leaves	0.70	60.5	204	[216]
Lauraceae	*Ocotea*	*splendens* (Meisn.) Baill	Brazil	NO	leaves	0.35	51.0	205	[217]
Lauraceae	*Persea*	*americana* Mill.	Nigeria	YES	leaves	0.20	43.9	216	[218]
Lauraceae	*Phoebe*	*porphyria* (Griseb.) Mez.	Argentina	NO	aerial parts	0.15	19.3	222	[219]
Magnoliaceae	*Magnolia*	*obovata* Thunb.	Japan	NO	leaves	0.05	23.7	173	[220]
Malvaceae	*Pachira*	*glabra* Pasq.	Nigeria	NO	leaves	0.71	14.5	212	[221]
Malvaceae	*Triumfetta*	*rhomboidea* Jacq.	Burkina-Faso	NO	aerial parts	0.02	24.2	282	[222]
Meliaceae	*Aglaia*	*odorata* Lour.	Thailand	NO	stem	0.07	10.2	15	[223]
Meliaceae	*Aphanamixis*	*polystachya* (Wall.) R.Parker	Bangladesh	NO	wood	n.a.	19.4	35	[224]
Meliaceae	*Cedrela*	*fissilis* Vellozo	Brazil	NO	leaves	0.06	26.3	70	[225]
Meliaceae	*Guarea*	*macrophylla* Vahl. *ssp. tuberculata* Vellozo	Brazil	NO	leaves	0.15	10.0	137	[226]
Moraceae	*Ficus*	*benjamina* L.	Nigeria	NO	leaves	n.a.	17.0	124	[227]
Myricaceae	*Morella*	*pensylvanica* (Mirbel) Kartesz	Canada	NO	aerial parts	0.15	14.5	190	[228]
Myristicaceae	*Gymnacranthera*	*canarica* (King) Warb.	India	NO	leaves	0.01	23.4	138	[229]
Myristicaceae	*Knema*	*kunstleri* Warb.	Malaysia	NO	aerial parts	0.12	23.2	159	[230]
Myristicaceae	*Myristica*	*malabarica* Lam.	India	NO	leaves	0.05	27.3	197	[229]
Myrtaceae	*Blepharocalyx*	*salicifolius* O.Berg	Brazil	NO	leaves	0.90	22.9	55	[231]
Myrtaceae	*Eucalyptus*	*leptophleba* F. Muell.	Australia	NO	leaves	0.01	11.4	118	[232]
Myrtaceae	*Eugenia*	*stipitata* McVaugh *ssp. sororia*	Portugal	NO	leaves	0.35	22.7	119	[233]
Myrtaceae	*Feijoa*	*sellowiana* Berg.	France	NO	fruits	0.10	12.0	121	[234]
Myrtaceae	*Marlierea*	*silvatica* Kiaersk	Brazil	NO	leaves	0.30	25.4	174	[235]
Myrtaceae	*Melaleuca*	*sphaerodendra var. microphylla* (Virot) Craven and J.W. Dawson	New Caledonia	NO	leaves	0.10	28.8	178	[236]
Myrtaceae	*Myrcia*	*cuprea* (O. Berg) Kiaersk.	Brazil	NO	aerial parts	0.10	39.1	194	[237]
Myrtaceae	*Myrcianthes*	*pseudo-mato (Legr.) Mc. Vaugh*	Argentina	NO	leaves	0.30	18.9	195	[238]
Myrtaceae	*Myrciaria*	*tenella* (DC.) Berg	Brazil	NO	leaves	0.40	25.1	196	[239]
Myrtaceae	*Ochrosperma*	*lineare* (C.T. White) Trudgen	Australia	NO	aerial parts	0.30	11.6	202	[240]
Myrtaceae	*Plinia*	*edulis* (Vell.) Sobral	Brazil	NO	leaves	0.10	21.2	235	[241]
Myrtaceae	*Psidium*	*striatulum* DC.	Brazil	NO	leaves	0.10	28.6	243	[242]
Myrtaceae	*Syzygium*	*aromaticum* L.	Morocco	YES	buds	8.58	27.5	270	[243]
Myrtaceae	*Syzygium*	*grande* (Wight) Walp.	Vietnam	NO	stem	0.12	29.3	271	[244]
Myrtaceae	*Uromyrtus*	*australis A. J. Scott*	Australia	NO	leaves	0.12	20.7	283	[245]
Papilionaceae	*Meristotropis*	*xanthioides* Vassilez	Iran	NO	aerial parts	3.20	11.8	184	[246]
Phyllanthaceae	*Actephila*	*excelsa* (Dazl.) Muell.	Vietnam	NO	leaves	0.15	11.2	7	[247]
Pinaceae	*Abies*	*nephrolepis* (Khingan fir)	South Korea	NO	needles	0.40	10.8	1	[248]
Pinaceae	*Pinus*	*pinaster* Aiton	Morocco	YES	needles	0.38	22.2	225	[249]
Pinaceae	*Pinus*	*armandii* Franch.	Scotland	NO	needles	n.a.	36.3	226	[250]
Pinaceae	*Pinus*	*bungeana* Zucc.	South Korea	NO	needles	0.31	27.2	227	[251]
Pinaceae	*Pinus*	*halepensis* Mill.	Turkey	NO	needles	n.a.	25.9	228	[252]
Piperaceae	*Piper*	*tuberculatum var. tuberculatum* (Micq.) CDC	Brazil	NO	leaves	n.a.	26.3	229	[253]
Piperaceae	*Piper*	*guineense* Schumach. and Thonn.	Cameroon	NO	seeds	1.1	57.6	230	[254]
Piperaceae	*Piper*	*nigrum* L.	India	YES	seeds	n.a.	45.3	231	[255]
Piperaceae	*Piper*	*maingayi* Hk. F.	Malaysia	NO	seeds	0.21	39.6	232	[256]
Piperaceae	*Pothomorphe*	*peltata* (L.) Miq.	Brazil	NO	leaves	0.20	68.0	239	[257]
Plantaginaceae	*Adenosma*	*indianum* (Lour.) Merr.	China	NO	aerial parts	0.29	10.32	8	[258]
Podocarpaceae	*Afrocarpus*	*mannii* (Hook.f.) C.N.Page	S. Tomé e Principe	NO	leaves	0.15	13.1	12	[259]
Ptaeroxylaceae	*Cedrelopsis*	*grevei* H. Baillon	Madagascar	NO	barks	n.a.	10.6	71	[260]
Rosaceae	*Agrimonia*	*eupatoria* L.	Iran	YES	flowers	1.20	42.8	16	[261]
Rosaceae	*Rosa*	*canina* L.	Tunisia	YES	flowers	1.40	32.0	250	[262]
Rubiaceae	*Cruciata*	*laevipes* Opiz	Italy	YES	aerial parts	0.70	19.0	97	[263]
Rubiaceae	*Geophila*	*repens* (L.) I.M. Johnst	China	NO	aerial parts	0.07	23.3	129	[264]
Rutaceae	*Aegle*	*marmelos* (L.) Corr.	Nepal	YES	leaves	0.29	29.6	10	[265]
Rutaceae	*Amyris*	*elimifera* L.	Cuba	NO	leaves	0.60	37.8	28	[266]
Rutaceae	*Atalantia*	*buxifolia* (Poir.) Oliv.	China	NO	leaves	0.36	25.8	47	[267]
Rutaceae	*Boenninghausenia*	*albiflora* Reichb.	India	NO	flowers	0.20	13.1	56	[268]
Rutaceae	*Citrus*	*garrawayi* F.M.Bailey	Australia	NO	leaves	0.20	17.6	80	[269]
Rutaceae	*Feroniella*	*lucida* (Scheff.) Swing	Thailand	NO	leaves	0.12	26.6	122	[270]
Rutaceae	*Flindersia*	*pimenteliana* F.Muell.	Australia	NO	leaves	0.03	16.9	126	[271]
Rutaceae	*Haplophyllum*	*villosum* (M. B.) G. Don	Iran	NO	aerial parts	0.22	13.1	139	[272]
Rutaceae	*Medicosma*	*obovata* T.G. Hartley	Australia	NO	aerial parts	0.40	17.2	177	[273]
Rutaceae	*Melicope*	*peninsularis* T.G. Hartley	Australia	NO	leaves	0.10	49.0	179	[274]
Rutaceae	*Murraya*	*paniculata* L.	Brazil	NO	leaves	0.03	57.6	192	[275]
Rutaceae	*Murraya*	*koenigii* (L.) Spreng	India	YES	leaves	0.1	45.9	193	[276]
Rutaceae	*Pamburus*	*missionis* (Wight) Swingle	India	NO	leaves	0.05	25.4	213	[277]
Rutaceae	*Spiranthera*	*odoratissima* A. St. Hil.	Brazil	NO	leaves	n.a.	23.8	267	[278]
Rutaceae	*Zanthoxylum*	*veneficum* F.M.Bailey	Australia	NO	leaves	0.10	36.3	292	[279]
Sapindaceae	*Acer*	*truncatum* Bunge	China	NO	leaves	n.a.	12.9	3	[280]
Schisandraceae	*Kadsura*	*coccinea* (Lem.) A.C. Smith	China	NO	roots	0.20	24.9	157	[281]
Scrophulariaceae	*Buddleia*	*asiatica* Lour.	India	NO	leaves	0.30	15.8	58	[282]
Scrophulariaceae	*Capraria*	*biflora* L.	Brazil	NO	leaves	0.09	29.6	66	[283]
Solanaceae	*Solanum*	*stipulaceum* Roem and Schult	Brazil	NO	flowers	0.08	25.8	265	[284]
Verbenaceae	*Aloysia*	*virgata* Juss.	Cuba	NO	aerial parts	n.a.	15.4	22	[285]
Verbenaceae	*Lantana*	*montevidensis* Briq	Brazil	NO	leaves	0.13	31.5	163	[286]
Verbenaceae	*Lantana*	*camara* L.	Madagascar	NO	aerial parts	0.08	43.61	164	[287]
Verbenaceae	*Lippia*	*myriocephala* Schltdl. et Cham.	Costa Rica	NO	leaves	0.08	16.1	171	[288]
Verbenaceae	*Petitia*	*domingensis* Jacq.	Cuba	NO	flowers	n.a.	35.7	219	[289]
Zingiberaceae	*Aframomum*	*corrorima* (Braun) P.C.M. Jansen	Ethiopia	NO	leaves	0.50	60.7	11	[290]
Zingiberaceae	*Alpinia*	*purpurata* (Viell.)	Fiji	NO	flowers	0.05	24.2	23	[291]
Zingiberaceae	*Alpinia*	*conchigera* Griff.	Malaysia	NO	rhizomes	0.14	10.0	24	[292]
Zingiberaceae	*Alpinia*	*mutica* Roxb.	Vietnam	NO	fruit	0.17	22.6	25	[293]
Zingiberaceae	*Alpinia*	*pinnanensis* T. L. Wu and Senjen	Vietnam	NO	fruit	0.23	11.4	26	[294]
Zingiberaceae	*Costus*	*afer* Ker–Grawl	Nigeria	NO	leaves	n.a.	12.3	93	[295]
Zingiberaceae	*Curcuma*	*longa* L.	India	YES	rhizomes	2.20	9.8	99	[296]
Zingiberaceae	*Etlingera*	*elatior* (Jack) R. M. Smith	Malaysia	NO	leaves	0.70	10.7	117	[297]
Zingiberaceae	*Globba*	*schomburgkii* Hook. f.	India	NO	aerial parts	0.01	31.7	133	[298]
Zingiberaceae	*Hedychium*	*coronarium* Koen.	Brazil	YES	leaves	0.68	43.0	140	[299]
Zingiberaceae	*Renealmia*	*breviscapa* Poepp. and Endl.	Brazil	NO	rhizomes	0.01	62.3	247	[300]
Zingiberaceae	*Renealmia*	*alpinia* (Rottb.) Maas	Brazil	NO	leaves	0.50	22.9	248	[301]
Zingiberaceae	*Zingiber*	*nimmonii* Dalzell	India	NO	rhizomes	0.04	42.2	293	[302]

**Table 2 ijms-21-06540-t002:** General statistics on BCP and yield percentages of plant species listed in Table 1.

Specification	Essential Oil Yield	Percentage of BCP
Number of cases	243	295
Range		
Minimum	0.00	9.8
Maximum	8.58	75.6
Mean	0.42	27.4
S.E.M.	0.06	0.8
S.D.	0.87	13.6
C.V. %	2.09	0.5

S.E.M., standard error of the mean; S.D., standard deviation; C.V., coefficient of variation.

**Table 3 ijms-21-06540-t003:** Average percentages of BCP and yields from plant parts as reported in plant species listed in Table 1. (±S.E.M.); n.c., not computable; E.O., essential oil.

Plant Part	Number of Species	BCP %	E.O. Yield %
Aerial Parts	115	25.19 (±1.10)	0.42 (±4.85)
Barks	3	39.03 (±18.59)	0.30 (±0.22)
Branches	1	34.90 (±n.c.)	0.20 (± n.c.)
Buds	1	27.50 (± n.c.)	8.58 (± n.c.)
Flowers	16	29.29 (±3.11)	0.41 (±0.13)
Fruits	9	26.93 (±4.43)	0.24 (±0.07)
Leaves	128	27.58 (±1.15)	0.30 (±0.04)
Oleo-gum resin	4	66.13 (±4.54)	15.50 (±8.30)
Rhyzomes	5	27.38 (±10.65)	0.49 (±0.43)
Roots	7	39.77 (±5.37)	1.77 (±0.92)
Stems	3	25.07 (±7.66)	0.13 (±0.04)
Twigs	2	16.25 (±4.85)	0.02 (±0.01)
Wood	1	19.40 (± n.c.)	0.42 (±n.c.)

**Table 4 ijms-21-06540-t004:** Average percentages of BCP and yields from plant families belonging to the plant species reported in Table 1. (±S.E.M.); n.c., not computable; n.a., not available; E.O., essential oil.

Family	Number of Species	BCP%	E.O. Yield%
Anacardiaceae	2	13.25 (±2.65)	n.a.
Annonaceae	15	22.17 (±1.26)	0.20 (±0.05)
Apiaceae	16	30.96 (±4.15)	0.37 (±0.14)
Apocynaceae	3	17.63 (±3.05)	0.26 (±0.10)
Araliaceae	1	39.00 (n.c.)	0.04 (n.c.)
Aristolochiaceae	2	26.65 (±3.75)	0.21 (±0.13)
Asteraceae	50	27.94 (±1.92)	0.47 (±0.14)
Atherospermaceae	1	32.20 (n.c.)	0.06 (n.c.)
Boraginaceae	2	22.95 (±10.15)	0.15 (±0.10)
Burseraceae	5	24.20 (±4.83)	0.14 (±0.02)
Cannabaceae	3	20.24 (±5.14)	0.27 (±0.14)
Caryophyllaceae	1	46.60 (n.c.)	0.48 (n.c.)
Cephalotaxaceae	1	41.60 (n.c.)	0.82 (n.c.)
Clusiaceae	5	25.85 (±6.84)	0.29 (±0.19)
Convolvulaceae	1	15.10 (n.c.)	n.a.
Cupressaceae	3	23.83 (±9.60)	1.59 (±0.84)
Cyperaceae	1	38.40 (n.c.)	n.a.
Ehretiaceae	2	41.95 (±15.65)	1.10 (n.c.)
Euphorbiaceae	6	25.60 (±15.42)	0.42 (±0.46)
Fabaceae	11	36.92 (±6.15)	3.89 (±3.45)
Flacourtiaceae	2	27.75 (±3.15)	n.a.
Geraniaceae	1	13.10 (n.c.)	0.22 (n.c.)
Gramineae	2	27.90 (±13.50)	0.19 (±0.09)
Hernandiaceae	1	9.80 (n.c.)	2.20 (n.c.)
Hypericaceae	3	41.10 (±15.86)	0.13 (±0.05)
Juglandaceae	1	10.00 (n.c.)	0.15 (n.c.)
Lamiaceae	57	31.03 (±2.03)	0.41 (±0.17)
Lauraceae	13	29.33 (±3.14)	0.38 (±0.18)
Magnoliaceae	1	56.90 (n.c.)	0.30 (n.c.)
Malvaceae	2	19.70 (±5.20)	0.11 (±0.04)
Meliaceae	4	30.55 (±9.27)	0.14 (±0.03)
Moraceae	1	37.80 (n.c.)	0.60 (n.c.)
Myricaceae	1	18.10 (n.c.)	0.30 (n.c.)
Myristicaceae	3	42.93 (±10.61)	1.35 (±0.85)
Myrtaceae	15	23.49 (±2.17)	0.27 (±0.08)
Papilionaceae	1	52.00 (n.c.)	0.30 (n.c.)
Phyllanthace	1	10.70 (n.c.)	n.a.
Pinaceae	5	23.22 (±5.33)	0.20 (±0.06)
Piperaceae	5	19.70 (±2.26)	0.23 (±0.07)
Plantaginaceae	1	20.90 (n.c.)	5.00 (n.c.)
Podocarpaceae	1	12.90 (n.c.)	n.a.
Ptaeroxylaceae	1	11.30 (n.c.)	0.72 (n.c.)
Rosaceae	2	18.00 (±6.60)	0.10 (±0.08)
Rubiaceae	2	17.15 (±0.25)	0.03 (n.c.)
Rutaceae	15	22.97 (±2.69)	0.27 (±0.06)
Sapindaceae	1	36.30 (n.c.)	n.a.
Schisandraceae	1	32.00 (n.c.)	1.40 (n.c.)
Scrophulariaceae	2	21.75 (±0.65)	0.10 (n.c.)
Solanaceae	1	12.20 (n.c.)	0.20 (n.c.)
Verbenaceae	5	24.70 (±6.58)	1.59 (±1.20)
Zingiberaceae	13	28.61 (±4.25)	0.22 (±0.06)

**Table 5 ijms-21-06540-t005:** Average percentages of BCP and yields from countries from which plant species reported in Table 1 were sampled. (±S.E.M.); n.c., not computable; n.a., not available; E.O., essential oil.

Country	Number of Species	BCP%	E.O. Yield%
Algeria	3	20.67 (±4.65)	0.61 (±0.59)
Argentina	8	25.85 (±5.41)	0.19 (±0.07)
Australia	18	25.70 (±2.98)	0.18 (±0.04)
Austria	2	34.85 (±18.65)	0.15 (n.c.)
Bangladesh	1	19.40 (n.c.)	n.a.
Benin	3	43.77 (±17.44)	0.09 (±0.01)
Brazil	56	33.01 (±2.20)	1.08 (±0.59)
Burkina Faso	2	21.65 (±2.55)	0.38 (±0.36)
Cameroon	4	34.40 (±8.12)	0.56 (±0.30)
Canada	1	14.50 (n.c.)	0.15 (n.c.)
Chile	1	21.10 (n.c.)	0.15 (n.c.)
China	11	19.26 (±2.54)	0.18 (±0.04)
Colombia	2	26.45 (±5.75)	0.11 (±0.05)
Costa Rica	3	19.53 (±5.61)	0.16 (±0.07)
Croatia	1	35.40 (n.c.)	0.35 (n.c.)
Cuba	5	36.30 (±10.85)	0.41 (±0.12)
Egypt	2	36.95 (±12.95)	2.30 (±0.30)
Ethiopia	2	41.15 (±19.55)	0.32 (±0.19)
Fiji	1	24.20 (n.c.)	0.05 (n.c.)
France	2	12.35 (±0.35)	0.15 (±0.05)
French Guian	1	47.00 (n.c.)	0.06 (n.c.)
Greece	1	16.40 (n.c.)	1.00 (n.c.)
India	29	27.00 (±2.32)	0.34 (±0.11)
Iran	30	28.69 (±2.02)	0.67 (±0.22)
Italy	5	22.54 (±2.55)	0.26 (±0.12)
Ivory Coast	5	22.24 (±3.48)	0.25 (±0.13)
Jamaica	1	21.70 (n.c.)	0.03 (n.c.)
Japan	3	20.93 (±1.94)	0.05 (±0.02)
Kenya	3	24.97 (±2.94)	0.17 (±0.12)
Lebanon	2	31.55 (±18.65)	0.20 (±0.11)
Lithuania	1	14.20 (n.c.)	0.05 (n.c.)
Madagascar	3	29.10 (±9.74)	0.14 (±0.06)
Malaysia	7	20.56 (±3.98)	0.25 (±0.11)
Martinique	1	21.10 (n.c.)	0.11 (n.c.)
Morocco	2	24.85 (±2.65)	4.48 (±4.10)
Nepal	1	29.60 (n.c.)	0.29 (n.c.)
New Caledonia	1	28.80 (n.c.)	0.10 (n.c.)
Niger	1	24.00 (n.c.)	0.30 (n.c.)
Nigeria	10	25.87 (±4.39)	0.19 (±0.09)
North Korea	1	24.00 (n.c.)	0.40 (n.c.)
Pakistan	2	21.40 (±3.90)	0.03 (n.c.)
Portugal	2	23.65 (±0.95)	0.26 (±0.09)
S. Tomé e Prince	1	13.10 (n.c.)	0.15 (n.c.)
Scotland	1	36.30 (n.c.)	n.a.
Serbia	4	18.35 (±3.47)	0.05 (±0.03)
South Africa	5	24.86 (±3.65)	0.09 (±0.02)
South Korea	4	16.30 (±3.83)	0.36 (±0.05)
Spain	4	28.23 (±4.44)	1.23 (±0.93)
Sri Lanka	1	37.00 (n.c.)	0.50 (n.c.)
Thailand	2	18.40 (±8.20)	0.10 (±0.03)
Togo	1	25.20 (n.c.)	0.50 (n.c.)
Tunisia	3	32.00 (±5.08)	0.52 (±0.44)
Turkey	14	29.21 (±3.51)	0.25 (±0.08)
USA	2	47.45 (±25.45)	2.10 (n.c.)
Venezuela	2	36.40 (±9.30)	0.06 (±0.01)
Vietnam	11	22.38 (±4.01)	0.28 (±0.08)

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
