# Peer review of "Plant Natural Sources of the Endocannabinoid (E)-β-Caryophyllene: A Systematic Quantitative Analysis of Published Literature"

_ijms, 2020, doi:10.3390/ijms21186540_

Round 1
Reviewer 1 Report
Dear authors,
The manuscript addresses an issue that is very important and relevant. However, this manuscript is not ready for publication yet. There are a few issues that the authors need to address before the manuscript can be considered for publication:
Although the title of the manuscript highlights the endocannabinoid action of (E)-β-caryophyllene, this topic is not addressed in the results and discussion. Therefore, I suggest that the title be changed to suit the actual content of the article.
The bibliographic survey was well done and the analysis of the literature data allows easy understanding. However, in table 1 it would be better to use the botanical family instead of the genus. In "Species and Auth", the full name of the species must be used. In addition, it should make clear what kind of proportion the "E.O. yield %" values refer to: mass / mass, mass / volume, volume / volume.
Author Response
I thank the reviewer for the comments and suggestion on the manuscript.
About the tile, I would prefer to keep the term endocannabinoid in the title. There are no indication of the action in the title although the introduction describes the importance of the endocannabinoid system and the potential of the molecule.
Table 1 has been revised according to the reviewer's comment and now the plant species are ordered according to the plant family. As a consequence the reference numbers have been updated both in the table and the text. Table 1 also has the indication that the E.O. percentage is calculated on a volume/weight basis.
In "Species and Auth", the full name of the species has been always used. I would be happy to edit in case the reviewer has spotted a species with incomplete full name. If the reviewer refers to the author name, I used the international system as indicated by IPNI.
I hope that the changes made to the manuscript may be accepted by the reviewer.
Reviewer 2 Report
The paper provides a systematic analysis of scientific literature to identify the plants with the highest (E)-β-caryophyllene and yield percentages production.
The paper is important to support the chemical extraction of this cannabinoid able to interact specifically with the CB2 receptor.
I have no particular changes to suggest to the author. For me the paper is acceptable for the publication it stands.
Author Response
I thank very much the reviewer for the kind comments